# Unraveling the Mechanism of Epichaperome Modulation by Zelavespib: Biochemical Insights on Target Occupancy and Extended Residence Time at the Site of Action

**DOI:** 10.3390/biomedicines11102599

**Published:** 2023-09-22

**Authors:** Sahil Sharma, Suhasini Joshi, Teja Kalidindi, Chander S. Digwal, Palak Panchal, Sang-Gyu Lee, Pat Zanzonico, Nagavarakishore Pillarsetty, Gabriela Chiosis

**Affiliations:** 1Chemical Biology Program, Memorial Sloan Kettering Cancer Center, New York, NY 10065, USAsuhasinij14@gmail.com (S.J.); digwalc@mskcc.org (C.S.D.); panchap1@mskcc.org (P.P.); 2Department of Radiology, Memorial Sloan Kettering Cancer Center, New York, NY 10065, USA; kalidint@mskcc.org (T.K.); lees3@mskcc.org (S.-G.L.); zanzonip@mskcc.org (P.Z.); 3Breast Cancer Medicine Service, Memorial Sloan Kettering Cancer Center, New York, NY 10065, USA

**Keywords:** zelavespib, epichaperomes, drug–target residence time, target–ligand interactions, target engagement and occupancy, mechanisms of drug action, icapamespib, drug discovery for cancers and neurodegenerative diseases

## Abstract

Drugs with a long residence time at their target sites are often more efficacious in disease treatment. The mechanism, however, behind prolonged retention at the site of action is often difficult to understand for non-covalent agents. In this context, we focus on epichaperome agents, such as zelavespib and icapamespib, which maintain target binding for days despite rapid plasma clearance, minimal retention in non-diseased tissues, and rapid metabolism. They have shown significant therapeutic value in cancer and neurodegenerative diseases by disassembling epichaperomes, which are assemblies of tightly bound chaperones and other factors that serve as scaffolding platforms to pathologically rewire protein–protein interactions. To investigate their impact on epichaperomes in vivo, we conducted pharmacokinetic and target occupancy measurements for zelavespib and monitored epichaperome assemblies biochemically in a mouse model. Our findings provide evidence of the intricate mechanism through which zelavespib modulates epichaperomes in vivo. Initially, zelavespib becomes trapped when epichaperomes bound, a mechanism that results in epichaperome disassembly, with no change in the expression level of epichaperome constituents. We propose that the initial trapping stage of epichaperomes is a main contributing factor to the extended on-target residence time observed for this agent in clinical settings. Zelavespib’s residence time in tumors seems to be dictated by target disassembly kinetics rather than by frank drug–target unbinding kinetics. The off-rate of zelavespib from epichaperomes is, therefore, much slower than anticipated from the recorded tumor pharmacokinetic profile or as determined in vitro using diluted systems. This research sheds light on the underlying processes that make epichaperome agents effective in the treatment of certain diseases.

## 1. Introduction

As in all fields of drug therapy, knowing the right molecular target and having a drug that perturbs that target is central to effective treatment, but it is not sufficient in itself. It is also necessary to know how and how long the target is being engaged and modulated in vivo at a therapeutic level [1,2,3,4]. 

Traditionally, in vivo potency estimates are projected from in vitro equilibrium drug–target measurements (k_d_, k_i_) to quantify the strength of drug–target interactions (i.e., affinity) combined with information on clearance from metabolic pathways [4,5,6,7]. In this model, drugs of high affinity for the target and commensurate systemic presence of the drug are needed to support a therapeutic effect in vivo. Recently, there has been growing interest in the concept of binding kinetics, wherein predictions regarding in vivo efficacy are based on the duration (or residence time) of the drug–target complex, rather than solely on its equilibrium binding affinity [8,9,10,11,12,13,14,15,16,17,18]. In essence, the drug–target residence time model posits that pharmacodynamic effects persist as long as the drug remains bound to its target and wane once the drug dissociates from the drug–target complex. This model has emerged as an explanatory framework for drugs that exhibit sustained pharmacodynamic effects even after the systemic drug concentration has decreased to subtherapeutic levels.

Among drugs with long residence times at the site of action are epichaperome binding agents, such as zelavespib and icapamespib [19]. Epichaperomes represent oligomeric structures comprised of tightly associated chaperones, co-chaperones, and various other factors [19,20,21]. It is important to differentiate epichaperomes from chaperones, which are ubiquitously expressed in all cell types, under both normal and pathological conditions. By contrast, epichaperomes are primarily localized within diseased cells and tissues [19,20,21]. In terms of function, epichaperomes diverge significantly from chaperones. They serve as pathological scaffolds, orchestrating widespread alterations in protein–protein interactions (PPIs) across the proteome rather than functioning as protein folders involved in protein synthesis and degradation pathways [19,20,21,22,23,24,25]. Comprehensive analyses of protein networks influenced by epichaperomes have revealed that the formation of these complexes disrupts the connectivity of numerous individual proteins crucial for maintaining disease-specific phenotypes within their specific contexts [23,24,26,27,28,29]. Targeting these disease-specific PPI network dysfunctions through epichaperome inhibition, epichaperome agents have emerged as promising drug candidates for various diseases, including cancer and neurodegenerative disorders [19].

The chaperone heat shock protein 90 (HSP90) and heat shock cognate 70 (HSC70) are central components of epichaperome assemblies, as demonstrated in several disorders, including Alzheimer’s disease, Parkinson’s disease, and cancer [23,24,26,27,29]. Therefore, HSP90 and HSC70 provide a common point of vulnerability for drug binding to and inhibition of epichaperomes and, in turn, for reversal of aberrant PPI networks that may be exploited for therapeutic use in these diseases [24,29,30]. Crucially, the chaperones in epichaperomes exist as conformational mutants specific to disease states, allowing targeted inhibition of pathologic forms while preserving chaperone functions in normal cells [21,25,30,31]. This unique property allows discrimination between pathologic and physiologic conformers and thus the development of epichaperome-specific inhibitors, such as PU-H71 (zelavespib) (binds HSP90 in epichaperomes) [26], PU-AD (icapamespib) (binds HSP90 in epichaperomes) [30], LSI137 (binds HSC70 in epichaperomes) [29], and PU-WS13 (binds glucose-regulated protein 94 (GRP94) in epichaperomes) [25,31]. These agents rapidly dissociate from chaperones in normal tissues while remaining at the site of action for days after administration despite rapid clearance from plasma and non-diseased tissues associated with a fast in vivo metabolism [25,26,30,32]. 

Epichaperome-targeting agents have been extensively evaluated and have shown therapeutic efficacy in various disease models, including cancer, neurodegenerative disorders, infection, and inflammation [24,26,28,29,32,33,34,35]. Particularly noteworthy is the successful translation of compounds such as zelavespib and icapamespib into clinical trials for the treatment of cancer and Alzheimer’s disease [32,36,37,38]. Radiolabeled versions of zelavespib and icapamespib have provided valuable information on the time these agents spend at the site of action in patients (i.e., in tumors and disease-afflicted brains) [30,32,36,39]. For example, ^124^I-labeled derivatives of zelavespib and icapamespib were used to detect and quantify epichaperomes in vivo through positron emission tomography (PET) imaging. Both agents contain an endogenous iodine, enabling labeling with a PET-compatible radionuclide, iodine-124. In mice and humans, real-time single-tumor pharmacokinetic (PK) measurements were enabled by PET using tracer amounts of ^124^I-radiolabeled agents co-injected with therapeutic doses of epichaperome drugs [32]. Clinical investigations have revealed that the duration of zelavespib’s presence within tumors exhibits substantial variability, with half-lives spanning from 24 to 100 h [32]. This variability has been attributed to the levels of epichaperome abundance: higher epichaperome levels are associated with prolonged zelavespib residence times in tumors [26,30]. Importantly, the degree of target occupancy, as assessed via PET imaging, displayed a significant correlation with the extent of the observed anti-tumor effects [30,32], suggesting that evaluations at the level of individual tumors could be used to optimize dose and schedule selection in epichaperome therapy. In no instance in human patients has the plasma PK correlated with the tumor retention profile and the pharmacodynamic effects noted for zelavespib [30,32,39], supporting the concept that plasma assays provide only limited insight into drug–target interactions and are unsuitable for epichaperome therapy development. Importantly, epichaperomes constitute only a fraction of the chaperone pool within diseased cells (e.g., approximately 5–35% in cancer cells) [26,27,29]. Despite the abundance of chaperones such as HSP90, which is present in nearly all mammalian cells and is estimated to comprise ~224–336 g in a mature human weighing 70 kg, constituting about 16% of the protein mass, epichaperome agents demonstrate specific and prolonged engagement with their target within the disease site. Thus, both zelavespib and icapamespib are examples of the growing number of the drugs where drug–target residence time is a main driver for the observed pharmacodynamic effects and, in turn, therapeutic efficacy. The factors determining the prolonged retention of these drugs in vivo remain, however, poorly understood.

Here, we used zelavespib as an epichaperome agent prototype and investigated the relationship between target and ligand seeking mechanistic insights into the long presence and associated biological effect reported for such agents at the site of action. 

## 2. Methods

### 2.1. Cell Lines and Culture Conditions

The MDA-MB-468 human breast cancer cell line [40] (RRID:CVCL_0419) was obtained from the American Type Culture Collection and cultured in Dulbecco’s Modified Eagle’s medium—high glucose (DME-HG) supplemented with 10% FBS, 1% L-glutamine, 1% penicillin, and streptomycin. Cells were authenticated using short tandem repeat profiling and tested for mycoplasma.

### 2.2. Mouse Models

All animal studies were conducted in compliance with Memorial Sloan Kettering Cancer Center (MSKCC)’s guidelines and under Institutional Animal Care and Use Committee approved protocols #05-11-024. Athymic nude mice (Hsd:Athymic Nude-*Foxn1^nu^*, female, 20–25 g, 5 weeks old; RRID:MGI:5652489) were obtained from Envigo (Indianapolis, IN, USA) and were allowed to acclimatize at the MSKCC vivarium for 1 week prior to implanting tumors. Mice were maintained in ventilated cages and provided with food and water ad libitum. All mice in all studies were observed for clinical signs at least once daily.

### 2.3. Reagents

Zelavespib (PU-H71), deuterated zelavespib (PU-H71-d_6_), and ^131^I-labeled icapamespib ([^131^I]-PU-AD) were synthesized and purified as previously reported [41,42,43].

### 2.4. Biodistribution Studies in Mice

MDA-MB-468 (RRID:CVCL_0419) tumor xenografts were established on the forelimbs of female athymic nude mice (Hsd:Athymic Nude-*Foxn1^nu^*; Envigo; 5 weeks old; RRID:MGI:5652489) by subcutaneous (s.c.) injection of 7 × 10^6^ cells in a 300 µL cell suspension of a 1:1 *v*/*v* mixture of PBS with reconstituted basement membrane (BD Matrigel^TM^, Collaborative Biomedical Products Inc., Bedford, MA, USA). Tumor progression was monitored using vernier calipers. For biodistribution studies, female athymic nude mice with MDA-MB-468 tumor xenografts (~200–300 mm^3^) on the forelimbs were injected intravenously in the tail vein with zelavespib (50 mg/kg (PU-H71.HCl formulated in 10 mM Phosphate Buffer, pH = ~6.4)) at different time points (0 (co-injection), 6, 24, and 48 h) prior to tracer administration. At designated time points post-drug administration, mice were injected intravenously in the tail vein with 0.93–1.1 MBq (25–50 µCi) of [^131^I]-PU-AD in 200 µL of 5% ethanol in saline. The activity in the syringe before and after administration was assayed in a dose calibrator (CRC-15R; Capintec, Florham Park, NJ, USA) to determine by difference the actual activity administered to each animal. Animals were euthanized by CO_2_ asphyxiation at 24 h post-administration of the radiolabeled [^131^I]-PU-AD tracer, and organs, including tumor(s), were harvested and weighed. ^131^I was measured in a 2480 WIZARD automatic gamma counter (PerkinElmer, Waltham, MA, USA) using a 260–430 keV energy window. Count data were background- and decay-corrected to the time of injection, and the percent injected dose per gram (%ID/g) for each tissue sample was calculated using a measured ^131^I calibration factor to convert count rate to activity and the activity was normalized to the activity injected and the weighed mass of the sample to yield the activity concentration in %ID/g.

### 2.5. LC–MS/MS Analyses of Tumors

Frozen tumors were weighed prior to homogenization in acetonitrile/H_2_O (3:7). Zelavespib was extracted in methylene chloride, and the organic layer was separated and dried under vacuum. Samples were reconstituted in mobile phase. Concentrations of drug were determined by high-performance LC–MS/MS. PU-H71-d_6_ was added as the internal standard [42]. Compound analysis was performed on the 6410 LC–MS/MS system (Agilent Technologies, Santa Clara, CA, USA) in multiple reaction monitoring (MRM) mode using positive-ion electrospray ionization. The following MRMs were monitored: PU-H71, 513/454, and 513/174; PU-H71-d_6_, 519/460, and 519/179. A Zorbax Eclipse XDB-C18 column (2.1 × 50 mm, 3.5 µm) was used for the LC separation, and the analyte was eluted under an isocratic condition (80% H_2_O + 0.1% HCOOH: 20% CH_3_CN) for 3 min at a flow rate of 0.4 mL/min.

### 2.6. Epichaperome Determination by Native PAGE

For epichaperome analysis, tumors were cut into small pieces and homogenized in native lysis buffer (20 mM HEPES pH 7.5, 50 mM KCl, 5 mM MgCl_2_, 0.01% NP40, 20 mM Na_2_MoO_4_) containing protease and phosphatase inhibitors using BioMasher^®^ II Micro Tissue Homogenizers (VWR, Radnor, PA, USA). Samples were left on ice for 30 min and then subjected to native PAGE analysis. Protein concentrations were determined using the BCA kit (Pierce, Waltham, MA, USA) according to the manufacturer’s instructions. Protein lysate (10–60 µg) was loaded onto 5.5% native gel and resolved at 4 °C, transferred to nitrocellulose membrane, and probed with the indicated primary antibodies: HSP90α (AB2928; RRID:AB_303423; 1:8000) from Abcam (Cambridge, UK), HOP (ADI-SRA-1500; RRID:AB_10618972; 1:1000) from Enzo and CDC37 (4793S; RRID:AB_10695539; 1:1000) from Cell Signaling (Danvers, MA, USA). Membranes were then incubated with a 1:5000 dilution of a peroxidase-conjugated corresponding secondary antibody. Detection was performed using the ECL Enhanced Chemiluminescence Detection System (Pierce™ ECL Western Blotting Substrate) according to the manufacturer’s instructions.

### 2.7. Use of Microdose [^131^I]-PU-AD Co-Injected with a Therapeutic Dose of Zelavespib or [^124^I]-PU-AD Injected Alone to Predict Tumor Molar Concentrations of Zelavespib

Iodine-124 is a ‘non-residualizing’ isotope (i.e., free iodine-124 washes out from cells), and therefore, data derived from positron emission tomography imaging of MDA-MB-468-bearing mice that were administered a tracer amount of [^124^I]-PU-AD can be used to determine real-time zelavespib concentrations delivered to tumors. Because positron emission tomography provides information on radioactivity and not molar concentrations, radioactivity data from Pillarsetty et al. [32] were transformed into tumor concentrations to calculate the time-dependent zelavespib concentrations in tumors for an administered dose of 50 mg/kg. The drug concentration (in μM) in the tumor water space, [Drug_tumor_], at the time *t* post-administration is therefore:$${\left[{Drug}_{tumor}\right]}_{t}{=Drug}_{dose}\cdot \frac{[A\mathrm{tumor}]t}{100\%}\cdot \frac{1}{W}\cdot \frac{1\times 10^{6}}{MW}\;µM$$where Drug_dose_ is the administered drug dose (in mg), [A_tumor_]*_t_* is the activity concentration (in % injected dose per milliliter, %ID/mL) of the surrogate radiotracer, W is the water space of the tissue (in mL/g), MW is the molecular weight of the drug (i.e., zelavespib), and the factor 1 × 10^6^ converts the concentration to μM. For the drug zelavespib and the surrogate radiotracer [^131^I]-PU-AD co-injected in a ratio of ~1 to 10,000, the drug itself is essentially the only significant form of the drug–surrogate combination in the tumor. Thus, for a co-administered dose of zelavespib (zelavespib_dose_ in mg) and a tracer amount of [^124^I]-PU-AD, the drug concentration in the tumor water space (using an average water space, W = 0.8 mL/g, and the molecular weight MW = 512) is:$$[{\mathrm{Zelavespib}}_{\mathrm{tumor}}{]}_{t}={\mathrm{Zelavespib}}_{\mathrm{dose}}\;\cdot \;\frac{[A\mathrm{tumor}]t}{100\%}\cdot 2.44\times 10^{3}\;µM$$

### 2.8. Statistics and Reproducibility

Statistics were performed and graphs were generated using Prism 9 software (GraphPad, Boston, MA, USA). Statistical significance was determined using ANOVA, as indicated. Means and standard errors are reported for all results unless otherwise specified. Effects achieving a 95% confidence interval (i.e., *p* < 0.05) were interpreted as statistically significant. No statistical methods were used to pre-determine sample sizes, but these are similar to those generally employed in the field. No samples were excluded from any analysis unless explicitly stated. 

## 3. Results

### 3.1. Experimental Design

To study the zelavespib–epichaperome interaction in vivo, we selected, for simplicity, an epichaperome-positive tumor model (e.g., subcutaneously xenografted MDA-MB-468 tumors in mice) [26]. Using whole-body PET imaging after the injection of ^124^I-labeled zelavespib, Pillarsetty et al. reported drug presence in MDA-MB-468 tumors up to 120 h post-single dose administration (~T_1/2_ = 60 h) [32]. Conversely, zelavespib was no longer observed in non-disease, non-metabolizing tissues after ~8 h, clearing rapidly from plasma, paralleled by a similar rapid clearance from normal tissues (i.e., chaperone-positive but epichaperome-negative). In these tumors, a clear relationship between tumor retention, pharmacodynamic effects, and anti-tumor activity was observed [32], supporting the notion that zelavespib was in residence with its target over the time it resided in these tumors.

Using the pharmacokinetic data reported by Pillarsetty et al. for this mouse model [32] and fitting the data to a one-phase exponential decay equation, we calculated the half-life of zelavespib in the plasma, lung, heart, muscle, bone, and brain to be 4.38, 3.37, 3.10, 5.03, 2.84, and 2.65 h, respectively (Figure 1a). These values are very much in agreement with those reported for zelavespib in human patients [32,44]. For example, one clinical study reported a mean terminal half-life (T_1/2_) in the plasma of 8.4 ± 3.6 h, with no dependency on dose level [44]. Contrasting plasma and normal tissues and organs, the distribution of zelavespib in tumors was best fitted using a bi-exponential model. After an initial rapid clearance phase (Phase 1: 0 to 8 h, k = 0.241 h^−1^; T_1/2_ = 2.87 h), a slow terminal clearance phase followed (Phase 2: T_1/2_ = 46.8 h; k = 0.015 h^−1^) (Figure 1a). The first phase thus reflects the clearance of systemic agent, while the second is attributable to specific tumor retention.

This mouse model, especially during Phase 2, is thus optimal for studying target-ligand interactions in vivo in an open, dynamic system (i.e., whole body) with little to no interference from systemic rebinding mechanisms (i.e., where there is no zelavespib in circulation or bound to other tissues). Critically, only the intact drug molecule is found in tumors [32], and is target-bound in Phase 2; thus, interference from metabolites is also minimal. Another crucial aspect to consider is the rapid distribution of zelavespib throughout the body. When administered via a bolus injection, it swiftly establishes a state of ‘equilibrium mixture’ between the drug and its target. After achieving this equilibrium state throughout the body, zelavespib then undergoes rapid clearance from the bloodstream, leading to a body-wide ‘dilution state.’ Therefore the bolus zelavespib injection in this mouse model closely emulates the classical method used for measuring dissociation constants (k_off_), whereby an equilibrium mixture comprising the ligand, receptor, and ligand–receptor complex is subjected to dilution, and the subsequent timeline of complex dissociation is monitored to determine new equilibrium concentrations of both the ligand and the receptor [45,46]. Thus, the tumor pharmacokinetics profile recorded in the Phase 2 interval may be a direct reflection of the in vivo kinetics of target–drug unbinding (k_off_ = 0.016 h^−1^; R_t_ = 1/k_off_ = 67.56 h; Figure 1b).

### 3.2. Tumor Zelavespib Molar Concentration Measurements

To test our hypothesis, we monitored the clearance rate of zelavespib from epichaperome-positive lesions following a single-dose bolus administration of 50 mg/kg (intravenously) (Figure 2a). Considering zelavespib’s long tumor residence time in this model, we analyzed intratumoral concentrations within a 30 h to 72 h post-drug administration window (Figure 2b). Mice were sacrificed at 30 h, 48 h, and 72 h post-injection, and tumors were harvested for analysis. Liquid chromatography tandem mass spectrometry (LC–MS/MS) with deuterated zelavespib as an internal standard was used to determine zelavespib concentrations in the tumors. At these time points, intratumoral micromolar concentrations of 2.57 ± 0.80, 1.59 ± 0.30, and 0.98 ± 0.30 were measured (using an average tumor water space, W = 0.8 mL/g) (Figure 2b). These concentrations closely match the reported active levels in this tumor model, confirming the presence of pharmacologically active drug levels in tumors from 30 h and beyond after a single dose of zelavespib.

### 3.3. Target Occupancy by Zelavespib

The presence of zelavespib in tumors does not directly confirm that it is target-bound. The drug could be trapped in non-tumor areas (i.e., stroma or necrotic tissue) or be intracellular but not target-bound. To test this notion and determine the percentile target occupied by each of these intratumoral concentrations of zelavespib, we performed studies in which tracer amounts (that is, non-pharmacologically perturbing amounts) of the radiolabeled epichaperome ligand [^131^I]-PU-AD (~6 ng/g which is ~1/10,000 of the therapeutic zelavespib amount) were administered to mice intravenously 24 h prior to tumor zelavespib measurements (Figure 2a). The ^131^I isotope has a half-life of 8 days [47,48], making it suitable for measurements of drugs with an extended tumor retention profile, such as that observed for zelavespib. It emits γ radiation and β particles; thus, γ-counting can be applied to detect and quantify the radioactive signal in tumors and tissues upon mouse sacrifice. Because both zelavespib and [^131^I]-PU-AD bind to the same target, the epichaperome, the more zelavespib that is present in the tumor and is target-bound, the lower the radioactivity signal is in the tumor. Indeed, we observed that as the intratumor concentration of zelavespib decreased, the radioactivity in the tumor proportionally increased (Figure 3a,b). 

According to this study, we estimated target occupancies of 38.8%, 24.9%, and 11.3% at 30 h, 48 h, and 72 h post-single dose injection of 50 mg/kg zelavespib. Accounting for background signal (i.e., subtracting the background radioactivity signal recorded in normal tissues or organs, such as the heart) we calculated similar target occupancies, 41.5%, 22.4%, and 3.5% at 30 h, 48 h, and 72 h, consistent with the notion that little, if any, zelavespib is present in normal tissues or organs, as reported [32]. Combined with prior pharmacodynamic and efficacy studies of 50 mg/kg PU-H71 in the MDA-MB-468 xenograft [32], these data propose that intratumoral zelavespib is target-bound, with approximately half to minimal target occupancy observed in the time interval of 30 to 72 h post-administration (Figure 3b). 

### 3.4. Modulation of Epichaperomes by Zelavespib

In cellular studies, it has been proposed that zelavespib, upon binding, becomes trapped inside the epichaperome assemblies, leading to the disassembly of epichaperomes into individual components [24,27,29]. However, this biochemical mechanism’s impact on the residence time of zelavespib at the site of action remains poorly understood in vivo. To investigate this, we analyzed the epichaperome target in tumors at 30 h and 72 h post-zelavespib administration using non-denaturing electrophoresis (native PAGE) [49] (Figure 4a–c).

As previously mentioned, epichaperomes are comprised of various tightly associated chaperones, co-chaperones, and scaffolding proteins, including HSP90, HSC70, HSP110, CDC37, and HOP [26,29,50]. Epichaperomes represent only a fraction of the total chaperone pool, constituting a relatively small portion when compared to the abundant chaperones (typically ranging from 5–35%, depending on the specific cancer cell line, with approximately 30% reported for MDA-MB-468 cells, as referenced [26,27,29]). Consequently, when epichaperome-positive tumor homogenates are subjected to analysis using native PAGE gels, a limited number of both distinct and less distinct high-molecular-weight species become apparent for the epichaperomes (which remain tightly bound and thus retained during Native PAGE) in addition to the prevalent primary band(s) characteristic of chaperones (as illustrated in Figure 4a, top). This difference arises from the dynamic and short-lived nature of folding chaperone complexes, causing them to disassemble when subjected to native PAGE. 

At the 30 h time point following zelavespib administration, we observed what appeared to be a temporary increase in the epichaperome signal on native PAGE (approximately a 2.29-fold increase compared to baseline). However, no corresponding alteration was noted in the total chaperone levels on SDS-PAGE (refer to Figure 4a, bottom). Preclinical studies [24,26,27,29,32] lend support to the idea that this signal increase is attributable to the trapping or stabilization of epichaperome components by ligands upon binding to the target, rather than an increase in epichaperome levels (as demonstrated in Figure 4a, which shows an increase in the levels of epichaperome-component chaperones but no increase in their overall levels).

By contrast, at 72 h post-zelavespib administration, we observed a decrease in the epichaperome signal on native PAGE (approximately 4.15-fold decrease compared to 30 h). Epichaperome levels at 72 h were thus found to be lower than the baseline levels (approximately 1.8-fold decrease), while no observable change was seen in the total chaperone levels on SDS-PAGE (Figure 4a,b). This suggests that the disassembly of the epichaperomes occurred by 72 h. Whether this disassembly is due to a direct drug allosteric effect, an indirect cellular mechanism following epichaperome inhibition, or a combination of both remains unknown.

On the basis of these findings, it can be inferred that approximately 24.09% of the baseline epichaperome target remains in tumors at 72 h post-zelavespib administration. This finding aligns with the value determined for target occupancy by zelavespib (approximately 11.3% occupied at 72 h). Hence, about 50% of the remaining intratumoral zelavespib at 72 h post-injection is target-bound, indicating that zelavespib’s target residence is influenced not only by drug–target unbinding kinetics (i.e., k_off_) but also by target disassembly kinetics (Figure 4c). In other words, both target disassembly and drug–target unbinding contribute to the observed residence time of zelavespib at the site of action, suggesting that the off-rate of zelavespib from epichaperomes is much slower than anticipated from the recorded tumor pharmacokinetic profile.

## 4. Discussion

The present study aimed to investigate the zelavespib–epichaperome interaction in an in vivo model and its impact on drug residence time at the site of action. Our findings shed light on the complex relationship between zelavespib binding, epichaperome disassembly, and drug target occupancy.

The pharmacokinetic analysis revealed that zelavespib displayed a prolonged presence in MDA-MB-468 tumors, with a half-life of approximately 60 h, consistent with prior clinical studies in humans [32]. Notably, zelavespib rapidly cleared from non-disease, non-metabolizing tissues within 8 h, and normal tissues with chaperone expression but lacking epichaperome presence [32]. To understand the mechanism underlying zelavespib’s target residence, we explored its interaction with epichaperomes in tumors. In vitro studies have previously proposed that zelavespib becomes trapped within the epichaperome assemblies, leading to their disassembly into individual components [24,27,29]. However, the in vivo implications of this biochemical mechanism remained elusive.

Our analysis using non-denaturing electrophoresis (native PAGE) confirmed the presence of epichaperomes in MDA-MB-468 xenograft tumors, marked by distinct high-molecular-weight species along with characteristic chaperone bands. The observed transient increase in epichaperome signal at 30 h post-zelavespib administration suggested the trapping or stabilization of epichaperome components upon drug binding. However, this increase did not correspond to an elevation in total chaperone levels, supporting the notion of specific ligand-induced trapping within the epichaperomes. By 72 h post-zelavespib administration, a decline in the epichaperome signal was evident, indicating the disassembly of epichaperomes in the tumor. While the exact mechanism behind this disassembly remains unknown, our findings suggest that it might be a result of zelavespib’s direct allosteric effect or an indirect cellular response following epichaperome inhibition. The quantification of target occupancy at 72 h post-administration revealed that approximately 24.09% of the baseline epichaperome target remained in the tumors. This value aligned with the estimated target occupancy of zelavespib at the same time point (approximately 11.3%). Importantly, with about 50% of the remaining intratumoral zelavespib found to be target-bound, both target disassembly kinetics and drug–target unbinding kinetics contribute to the observed drug residence time at the site of action.

The mechanisms proposed to rationalize the long residence times of drugs can vary depending on the specific drug–target interaction and the biological context [4,8,51,52,53,54,55,56]. One common mechanism involves covalent binding, in which the drug forms a covalent bond with its target, leading to a stable drug–target complex that persists over an extended period. Another mechanism is the slow dissociation of the drug from its target due to strong non-covalent interactions, such as hydrogen bonding or hydrophobic interactions. This slow off-rate allows the drug to remain bound to the target for an extended time, ensuring sustained pharmacological effects. Additionally, drugs with large molecular sizes or those that undergo slow metabolic degradation may exhibit prolonged residence times in target tissues. Furthermore, certain drugs may be sequestered or trapped within specific cellular compartments or structures, contributing to their extended retention at the site of action. Understanding these mechanisms can aid in the rational design of drugs with optimized pharmacokinetic properties, allowing for enhanced drug efficacy and potentially reducing dosing frequency, leading to improved patient outcomes and therapeutic success. 

The mechanism of residence time for zelavespib appears to be different from what has been observed for other drugs due to its unique interaction with the epichaperome target. Zelavespib’s ability to become trapped inside the epichaperome assemblies and induce their disassembly sets it apart from conventional drug–target interactions. While many drugs rely on covalent binding or strong non-covalent interactions to prolong their residence times [5,57,58,59], zelavespib’s specific binding and stabilization within the epichaperome complex lead to an extended target residence. This phenomenon allows zelavespib to remain associated with its target over a longer duration compared with other drugs. Moreover, the slow off-rate of zelavespib from the epichaperome contributes to its sustained presence in the tumor microenvironment, which is distinct from the clearance profiles of many other drugs. The intricacies of the epichaperome assembly, the unique dynamics of epichaperome disassembly, and the interplay between drug binding and target kinetics contribute to zelavespib’s long residence time at the site of action. Understanding these distinctive mechanisms provides valuable insights into zelavespib’s pharmacological behavior and highlights the need for targeted approaches in the development of drugs with extended residence times. The elucidation of these mechanisms may have implications beyond zelavespib, potentially guiding the design of future therapies aimed at harnessing unique drug–target interactions to achieve prolonged and effective drug residence in specific tissues or cell compartments.

## 5. Conclusions

Drugs with long residence times at their target sites have garnered increasing interest in the field of drug development [3,8,12,15,60,61,62,63,64,65]. These compounds exhibit prolonged binding to their respective targets, leading to sustained pharmacological effects and enhanced therapeutic efficacy. In the case of zelavespib, our study demonstrates a remarkably long residence time within epichaperomes in tumors, possibly due to its specific binding and trapping mechanism. Such prolonged drug residence not only allows for sustained target inhibition but may also limit the frequency of drug administration, potentially reducing the risk of side effects and improving patient compliance. Understanding the factors influencing drug residence time, such as target interactions and intracellular dynamics, is crucial for designing more effective and durable cancer therapies. The insights gained from our investigation of zelavespib’s interaction with the epichaperome highlight the importance of considering drug residence time as a key determinant of therapeutic success and provide valuable groundwork for the development of future treatments with extended drug residence times in their target tissues. This study has implications for the development of agents such as zelavespib and icapamespib in cancers and neurodegenerative diseases.

## Figures and Tables

**Figure 1 biomedicines-11-02599-f001:**
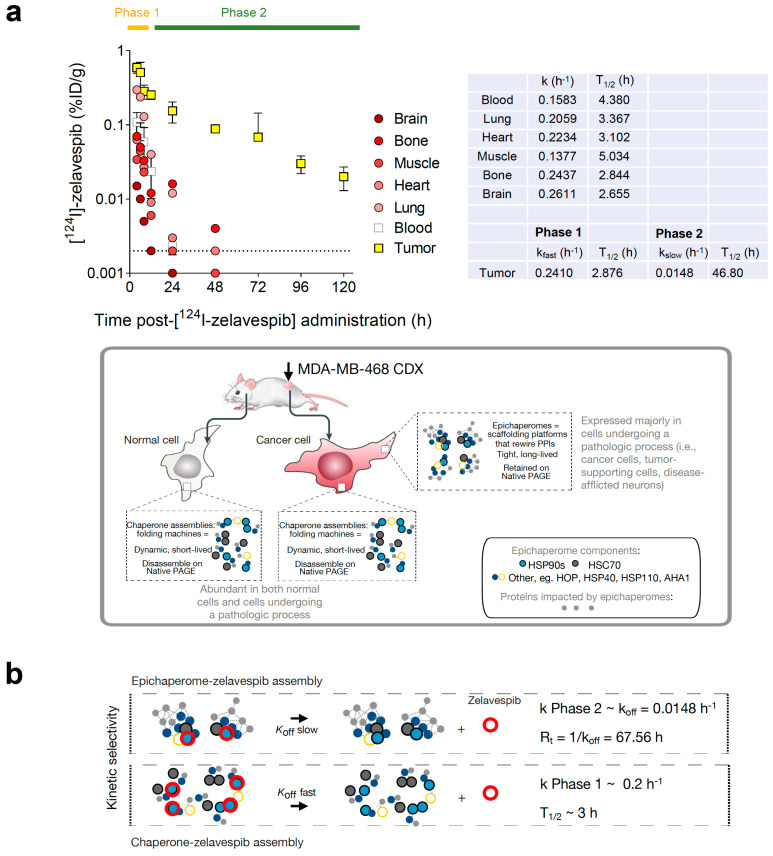
Biodistribution of zelavespib in MDA-MB-468 tumor-bearing mice. (**a**) Biodistribution of [^124^I]-Zelavespib in MDA-MB-468 tumors, organs, tissues, and plasma as per ref. [32]. Data are presented as means ± s.d., five mice per time point. Curves for blood, lung, heart, muscle, bone, and brain were fitted using a one-phase decay equation, and kinetics of zelavespib clearance were determined. Tumor curve was fitted to a two-phase exponential decay. k, k_fast_, and k_slow_ are the rate constants. Half-life (T_1/2_) is computed as ln(2)/k. Cartoon below shows the distinction between chaperones and epichaperomes in terms of their structure, function, and prevalence. (**b**) Illustrative schematic showing the distinct binding kinetics of zelavespib to epichaperomes (slow off-rate) and to chaperone assemblies (fast off-rate). The tumor pharmacokinetics profile recorded in the Phase 2 interval may represent a direct reflection of the in vivo kinetics of target–drug unbinding. k_off_, kinetic rate constant for dissociation of an epichaperome-zelavespib assembly. R_t_, retention time.

**Figure 2 biomedicines-11-02599-f002:**
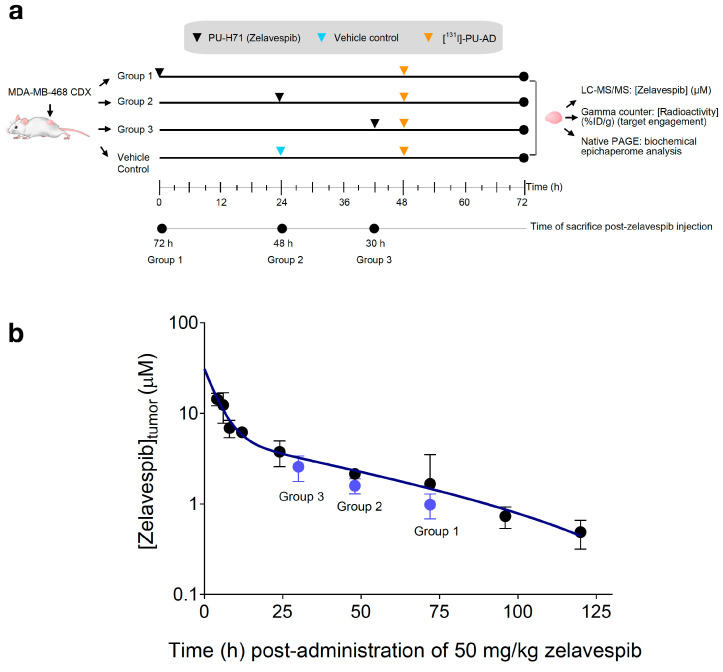
Molar concentration of zelavespib in MDA-MB-468 tumor-bearing mice following single-dose intravenous injection. (**a**) Experiment design to investigate the relationship between target and ligand seeking mechanistic insights into the long presence and associated biological effect reported for zelavespib at the site of action. (**b**) Tumor molar concentrations of zelavespib. Black circles, zelavespib tumor exposure calculated based on the mean %ID/g generated from the biodistribution curve from Figure 1a for an injected dose of 50 mg/kg. Blue circles, zelavespib molar concentrations experimentally determined by LC–MS/MS upon administration of 50 mg/kg zelavespib and sacrificing the mice at the indicated times post-zelavespib administration (*n* = 5 mice per time-point). Data are presented as means ± s.d.

**Figure 3 biomedicines-11-02599-f003:**
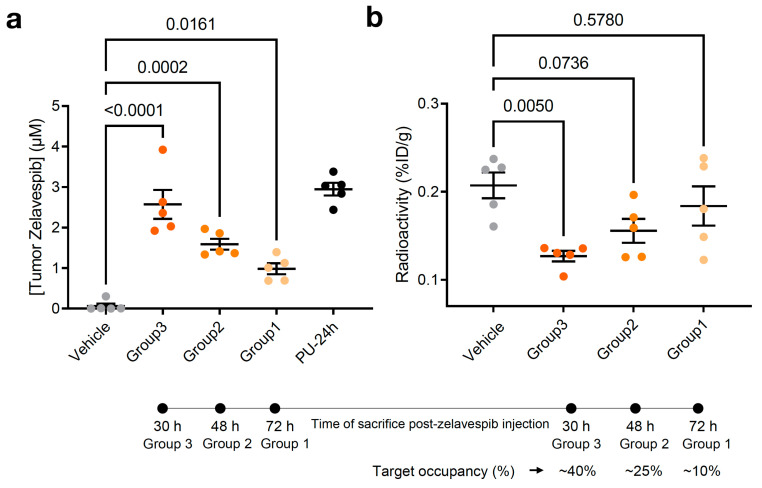
Relationship between molar concentration and target engagement in MDA-MB-468 tumor-bearing mice. (**a**) Tumor molar concentrations of zelavespib experimentally determined by LC–MS/MS upon administration of single-dose 50 mg/kg zelavespib and sacrificing the mice at 30 h, 48 h, and 72 h post-zelavespib administration (*n* = 5 mice per time-point). Data are presented as means ± s.e.m., *n* = 5, one-way ANOVA with Dunnett’s post hoc. Controls: PU-24h, mice were injected 50 mg/kg zelavespib, and tumor concentrations of the drug were determined at 24 h post-injection using LC–MS/MS. Vehicle, mice were injected vehicle only. (**b**) Target engagement by zelavespib was determined by co-injection of zelavespib with a trace amount of [^131^I]-PU-AD. Zelavespib was administered 6, 24, and 48 h prior to [^131^I]-PU-AD. Following mouse sacrifice at 24 h post [^131^I]-PU-AD injection, thus 30, 48, and 72 h post-zelavespib injection (see experimental design in Figure 2a), radioactivity was determined using a ɣ-counter to yield the activity concentration in %ID/g. Control: vehicle, mice were injected vehicle and [^131^I]-PU-AD only. Data are presented as means ± s.e.m., *n* = 5, one-way ANOVA with Dunnett’s post hoc. Experiment design in Figure 3a,b is as shown in Figure 2a.

**Figure 4 biomedicines-11-02599-f004:**
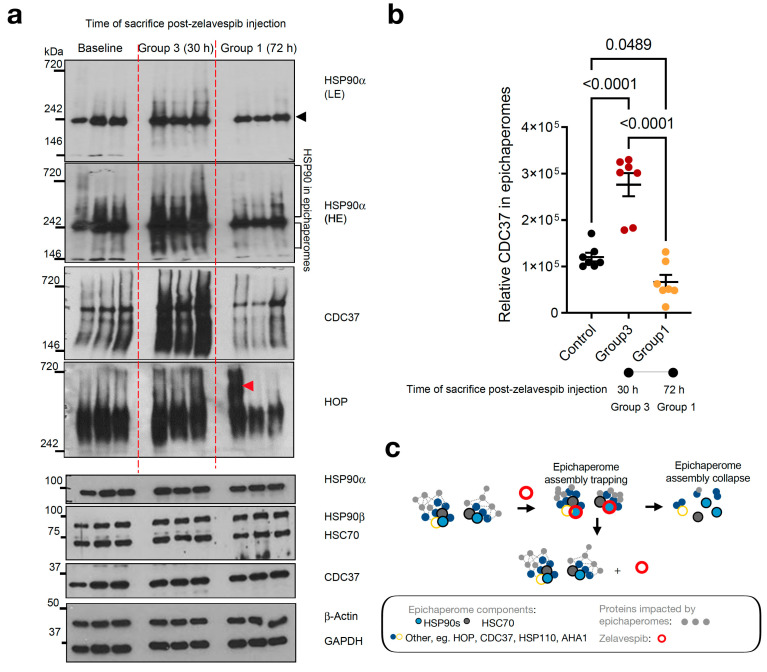
Epichaperome modulation by zelavespib in MDA-MB-468 tumor-bearing mice. (**a**) native-PAGE (top) or SDS-PAGE (bottom) followed by immunoblotting of epichaperome components (i.e., constituent chaperones and co-chaperones) in MDA-MB-468 tumors. Tumor-bearing mice were injected with a single dose of 50 mg/kg zelavespib, and tumor analysis was performed after sacrificing the mice at 30 h and 72 h post-zelavespib administration (*n* = 5 mice per time-point). Experiment design in Figure 4 is as shown in Figure 2a. Gels, representative tumors. To account for tumor heterogeneity (i.e., MDA-MB-468 are aggressive tumors with highly proliferative periphery adjacent to areas of necrosis), several cuts from the same tumor were analyzed when tumor size permitted. β-actin and GAPDH, protein loading controls; black arrowhead, biochemical signature of folding HSP90 pools as detected by native PAGE separation and immunoblotting; red arrowhead, non-specific signal, LE, low exposure; HE, high exposure. (**b**) Graph, data are presented as means ± s.e.m., *n* = 7, one-way ANOVA with Dunnett’s post hoc of tumors as in a. Because HSP90 in epichaperomes was not well separated from the folding HSP90 pool, we performed epichaperome quantification using CDC37 in epichaperomes as a surrogate marker. (**c**) Cartoon showing the proposed biochemical mechanism in vivo. Initially, zelavespib becomes trapped when epichaperomes bound, a mechanism that results in epichaperome disassembly, with no change in the expression level of epichaperome constituents. We propose that the initial trapping stage of epichaperomes is a main contributing factor to the extended on-target residence time observed for this agent in clinical settings. Zelavespib’s residence time in tumors seems to be dictated by target disassembly kinetics rather than by frank drug–target unbinding kinetics.

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
