# Peer review of "Unraveling the Mechanism of Epichaperome Modulation by Zelavespib: Biochemical Insights on Target Occupancy and Extended Residence Time at the Site of Action"

_biomedicines, 2023, doi:10.3390/biomedicines11102599_

Round 1

Reviewer 1 Report

 In the manuscript focus on epichaperome agents, such as zelavespib and icapamespib, which remain target bound for days after administration despite rapid clearance from plasma, non-diseased tissues also are metabolized fast. It is intresting. But the manuscript should to be proved to be accepted.  

1.      As an article, the abstract would to prove more data instead of using the word.

2.      In figure 1(b), why not there are only one group data such as ?

3.      In figure2 (b), why there only one blue dot for the Group1, 2, 3? How about the mean of the blank curve?

Minor editing of English language required.

Author Response

We sincerely thank the reviews for their thoughtful evaluation of the manuscript and for their insightful suggestions. We are also grateful all three reviewers have found numerous strengths in this work. Their support and recommendation for publication pending revision are invaluable to us. We look forward to sharing our research with the scientific community and contributing to the field.

We address below the comments and concerns of the reviewers. We also note here that more significant text modifications, including those in response to the editorial request, are highlighted using a blue font.

Review report (Round 1):

Reviewer 1:

Comments and Suggestions for Authors:

In the manuscript focus on epichaperome agents, such as zelavespib and icapamespib, which remain target bound for days after administration despite rapid clearance from plasma, non-diseased tissues also are metabolized fast. It is interesting. But the manuscript should be proved to be accepted.

  1. As an article, the abstract would prove more data instead of using the word.

Response: Thank you for your feedback and for recognizing the significance of our manuscript. We appreciate Reviewer#1’s suggestion to include specific data in the abstract. While we acknowledge that data inclusion can provide a glimpse of our research findings and enhance the abstract’s informativeness, the primary purpose of an abstract is to offer a concise summary of the study’s key aspects, including its objectives, methods, results, and implications. We believe that the current abstract effectively accomplishes this goal. The addition of specific data, while valuable in some cases, might risk making the abstract overly complex, a concern we hope Reviewer#1 would understand.

  1. In figure 1(b), why not there are only one group data such as?

Response: Thank you for noting the missing value. The schematic was updated by adding this value from Figure 1a also into Figure 1b.

  1. In figure 2(b), why there only one blue dot for the Group 1, 2, 3? How about the mean of the blank curve?

Response: Thank you for requesting this clarification. For the Vehicle control group (i.e., no zelavespib) the Molar concentration of zelavespib in tumors, as determined by LC-MS/MS, is graphed in Figure 3a. This shows that no zelavespib is detected in Vehicle only injected mice.

Comments on the Quality of English Language: Minor editing of English language required.

Reviewer 2 Report

Drugs with a long residence time at their target sites are often more efficacious in disease treatment. The mechanism, however, behind prolonged retention at the site of action is often difficult to understand for non-covalent agents. In this context, the authors took epichaperome agents, zelavespib and icapamespib as example, and investigated their impact on epichaperomes in vivo. Their findings sheds light on the underlying processes that make epichaperome agents effective in the treatment of certain diseases. This is an interesting manuscript. But I have several following concerns:

1. Abbreviations should be defined when they first appear in the text. Such as "PPIs"

2. all the "in vivo" and "in vitro" in the text should be italicized.

3. The formula font in the text should be the same as the other fonts in the text.

4. Please increase the resolution of the image in the text and make the image size suitable for the page.

5. It is recommended to re-run Figure 4a and quantify the relevant bands in 4a. In addition, the text in Figure 4c is too small to read clearly.

6. Please unify the format of references in the article, including the author's name, the case of words in the title of the article, the writing of the name of the journal, and the page number.

 Minor editing of English language required.

Author Response

We sincerely thank the reviews for their thoughtful evaluation of the manuscript and for their insightful suggestions. We are also grateful all three reviewers have found numerous strengths in this work. Their support and recommendation for publication pending revision are invaluable to us. We look forward to sharing our research with the scientific community and contributing to the field.

We address below the comments and concerns of the reviewers. We also note here that more significant text modifications, including those in response to the editorial request, are highlighted using a blue font.

Review report (Round 1):

Reviewer 2:

Comments and Suggestions for Authors:

Drugs with a long residence time at their target sites are often more efficacious in disease treatment. The mechanism, however, behind prolonged retention at the site of action is often difficult to understand for non-covalent agents. In this context, the authors took epichaperome agents, zelavespib and icapamespib as example, and investigated their impact on epichaperomes in vivo. Their findings shed light on the underlying processes that make epichaperome agents effective in the treatment of certain diseases. This is an interesting manuscript.

Response: Thank you for the positive assessment of our manuscript. Reviewer#2’s recognition of the significance of our research is greatly appreciated.

But I have several following concerns:

  1. Abbreviations should be defined when they first appear in the text. Such as "PPIs"

Response: Done as suggested.

  1. all the "in vivo" and "in vitro" in the text should be italicized.

Response: Done as suggested.

  1. The formula font in the text should be the same as the other fonts in the text.

Response: Done as suggested.

  1. Please increase the resolution of the image in the text and make the image size suitable for the page.

Response: We have revised the figure size and the font for each figure to make it suitable for the page, as suggested. High-resolution figures (in pdf, vector format) are provided, and are uploaded along the manuscript (which contains the low-resolution figures pasted into the text). The pdf should be used at the production stage to replace the low-res figures now pasted into the doc file. Thank you.

  1. It is recommended to re-run Figure 4a and quantify the relevant bands in 4a. In addition, the text in Figure 4c is too small to read clearly.

Response: To re-run Figure 4a, it would necessitate repeating the entire in vivo study. As indicated in the schematic in Figure 2a, detailed in the legend of Figure 4a, and outlined in the Methods section, each tumor underwent subdivision into multiple pieces. This division served two primary purposes: firstly, for the LC-MS/MS analyses to determine drug concentrations, and secondly, for native and SDS PAGE analyses aimed at quantifying epichaperomes and chaperones. It is crucial to note that MDA-MB-468 tumors, being highly aggressive, develop a necrotic core. Due to this characteristic, their size cannot surpass a certain threshold if we are to conduct reliable biochemical analyses. Therefore, we conducted as many measurements on these tumors as technically feasible, considering the aforementioned limitations.

We increased the font size in the Legend box of Figure 4c to improve readability, as suggested.

  1. Please unify the format of references in the article, including the author's name, the case of words in the title of the article, the writing of the name of the journal, and the page number.

Response: Done as suggested.

Comments on the Quality of English Language: Minor editing of English language required.

Reviewer 3 Report

This is an interesting and important  paper establishing the possible role of prolonged residence of Zelavespib on its target site of epichaperome. Thus, the paper considers important sublect and additionally is well written. Study is well scheduled and thought out. Thus, it should be published in present form. The small non-cler statement "....despite rapid clearance from plasma, non-diseased tissues also are metabolized fast." Should thetre be and non-dieased? could be corrected upon proof-reading

Author Response

We sincerely thank the reviews for their thoughtful evaluation of the manuscript and for their insightful suggestions. We are also grateful all three reviewers have found numerous strengths in this work. Their support and recommendation for publication pending revision are invaluable to us. We look forward to sharing our research with the scientific community and contributing to the field.

We address below the comments and concerns of the reviewers. We also note here that more significant text modifications, including those in response to the editorial request, are highlighted using a blue font.

Review report (Round 1):

Reviewer 3:

Comments and Suggestions for Authors:

This is an interesting and important paper establishing the possible role of prolonged residence of Zelavespib on its target site of epichaperome. Thus, the paper considers important subject and additionally is well written. Study is well scheduled and thought out. Thus, it should be published in present form.

Response: We extend our heartfelt gratitude for Reviewer#3’s positive and insightful assessment of our manuscript. His/her/their kind words and recognition of the significance of our work are truly appreciated. The encouraging feedback reinforces our commitment to advancing this important subject. We are pleased to learn that Reviewer#3 found the paper well-written and well-structured.

The small non-clear statement ".... despite rapid clearance from plasma, non-diseased tissues also are metabolized fast." Should there be and non-diseased? could be corrected upon proof-reading.

Response: We rectified the Abstract to clarify this sentence.

Round 2

Reviewer 2 Report

The authors have addressed all my concerns. I recommend accepting this manuscript in current status.